# The Effect of S-Allyl L-Cysteine on Retinal Ischemia: The Contributions of MCP-1 and PKM2 in the Underlying Medicinal Properties

**DOI:** 10.3390/ijms25021349

**Published:** 2024-01-22

**Authors:** Windsor Wen-Jin Chao, Howard Wen-Haur Chao, Hung-Fu Lee, Hsiao-Ming Chao

**Affiliations:** 1Department of Medicine, Aston Medical School, Aston University, Birmingham B4 7ET, UK; windsor.chao123@gmail.com; 2Department of Science, University of British Columbia, Vancouver, BC V6T 1Z4, Canada; wenhaur.chao@gmail.com; 3Department of Neurosurgery, Cheng Hsin General Hospital, Taipei 11220, Taiwan; ufae0073@ms7.hinet.net; 4Department of Chinese Medicine, School of Chinese Medicine, China Medical University, Taichung 40402, Taiwan; 5Department of Medicine, Institute of Pharmacology, School of Medicine, National Yang Ming Chiao Tung University, Taipei 11221, Taiwan; 6Department of Ophthalmology, Shin Kong Wu Ho-Su Memorial Hospital, Taipei 111045, Taiwan

**Keywords:** S-allyl L-cysteine, retinal pigment epithelium, retinal ganglion cell, pyruvate kinase M2, monocyte chemoattractant protein-1, hydrogen peroxide, oxidative stress, retinal ischemia

## Abstract

Retinal ischemia plays a vital role in vision-threatening retinal ischemic disorders, such as diabetic retinopathy, age-related macular degeneration, glaucoma, etc. The aim of this study was to investigate the effects of S-allyl L-cysteine (SAC) and its associated therapeutic mechanism. Oxidative stress was induced by administration of 500 μM H_2_O_2_ for 24 h; SAC demonstrated a dose-dependent neuroprotective effect with significant cell viability effects at 100 μM, and it concurrently downregulated angiogenesis factor PKM2 and inflammatory biomarker MCP-1. In a Wistar rat model of high intraocular pressure (HIOP)-induced retinal ischemia and reperfusion (I/R), post-administration of 100 μM SAC counteracted the ischemic-associated reduction of ERG b-wave amplitude and fluorogold-labeled RGC reduction. This study supports that SAC could protect against retinal ischemia through its anti-oxidative, anti-angiogenic, anti-inflammatory, and neuroprotective properties.

## 1. Introduction

Retinal ischemia plays an important role in vision-threatening retinal disorders, such as central or branch retinal artery/vein occlusion, age-related macular degeneration, glaucoma, etc. [1], which is largely linked with cumulative oxidative stress. These are pathological changes that occur at the cellular level [2]. For example, RPE is a type of tissue with limited regenerative ability, and its death can adversely affect rod and cone cells’ survival [3]. The progressive death of RPE cells could result in vision loss, which contributes to the development of retinal degenerative diseases, such as age-related macular degeneration. These patients usually present with impaired visual function and are unable to recognize facial features [4]. In the end, these adverse effects on vision might lead to depression and even mental breakdown. Of note, other cells like retinal ganglion cells (RGCs) also play an important role in retinal ischemia, as it is responsible for the transmission of light information from the retina to the brain [1]. In this case, the death of these neurons from retinal ischemia and reperfusion injury could also lead to pathological visual deterioration and blindness [1].

It is widely accepted that vascular endothelial growth factor (VEGF) is increasingly secreted during retinal ischemia. Hypoxia-inducible factor 1-alpha (HIF-1α) plays an important role in the upregulation of VEGF in cells subjected to oxidative stress [5]. Pyruvate kinase M2 (PKM2) is a co-activator of HIF-1α [6]. Thus, PKM2 and HIF-1α can influence each other in the activation of VEGF expression. In other words, PKM2 is able to physically and chemically stimulate HIF-1α, which results in VEGF secretion [6,7]. In addition to the development of fragile and permeable neovascular vessels prone to rupture, other aspects of critical consideration in the advent of ischemia are the inflammatory processes ensuing reperfusion. The blood reperfusion to ischemic tissues results in an exacerbated inflammatory response and subsequent retinal cell death [7]. 

Among the key molecular players in this context is monocyte chemoattractant protein-1 (MCP-1), which has been shown to induce retinal neovascularization. It plays a pivotal role in orchestrating the inflammatory cascade in ischemic retinopathy [8]. Specifically, MCP-1 is involved in regulating retinal neovascularization and inflammation caused by retina ischemia via increased chemotaxis involving cells like macrophages [8]. Furthermore, monocytes and macrophages have been reported to induce reactive oxygen species (ROS) production and inflammation within the body, eventually causing defined pathological processes of neovascularization, vascular permeability, and vascular dysfunction [9]. Importantly, PKM2 is shown to interact with MCP-1, which is demonstrated in the study of Doddapattar et al. (2022) [10]. PKM2-deficient mice is associated with a reduced activity of proinflammatory molecules, such as MCP-1, IL-12, and IL-1b [10]. The evidence shown above suggests that the overexpression of PKM2, HIF-1α, MCP-1, and VEGF biomarkers in retinal cells is largely correlated with the development of retinal ischemia-associated disorders, as described previously.

The aim of this study was to determine the mechanism of S-allyl L-cysteine (SAC), an active ingredient found within dry aged garlic extract that is said to possess anti-oxidative and neuroprotective effects [11]. In this case, garlic is a common food ingredient used among both Asians and Caucasians, and it has been presently selected for further evaluating its therapeutic effects. SAC’s anti-oxidant, anti-angiogenesis, and anti-inflammatory properties were determined through the cell culture of porcine RPEs under H_2_O_2_-induced oxidative stress and through an HIOP-induced retinal ischemia animal model. Anti-oxidants are often utilized as a treatment or prevention of ischemic-related disorders; thus early prevention and treatment with anti-oxidants is therefore an imperative consideration. In this case, various methods were utilized to observe different aspects, including cell viability, electrophysiologic function, and molecular assays (i.e., PKM2 and MCP-1). It is hypothesized that SAC is able to effectively and dose-dependently protect against oxidative stress and downregulate the level of PKM2 and MCP-1 via exhibiting its anti-oxidative, anti-ischemic, and anti-inflammatory effects. 

## 2. Results

### 2.1. Cell Death Ratio Was Evaluated Using an Optical Microscope

To observe the cell death rate, the number of stained (dead) or unstained RPE cells (living) was counted using epifluorescence microscopy and optical microscopy. Presently, various concentrations of SAC (25, 50, 100 μM) were added 15 min before the incubation of 500 μM H_2_O_2_ for 24 h. As shown in Figure 1, in the control group (A/F), RPE cells cultured in DMEM (32.48 ± 1.40%; *n* = 3) showed the least ratio of cell death relative to RPE cells subjected to 500 μM H_2_O_2_-induced oxidative stress (68.93 ± 1.25%; B/F; *n* = 3). In this case, the RPE cells subjected to 500 μM H_2_O_2_-induced oxidative stress demonstrated a significantly (* *p* < 0.05) greater cell mortality rate. In contrast, Figure 1B-F indicate that 15 min of pre-incubation with 25 (55.28 ± 0.77%; *n* = 3), 50 (54.96 ± 1.62%; *n* = 3), and 100 μM SAC (46.74 ± 2.20%; *n* = 3) revealed a dose-dependent and significant (100 μM; * *p* < 0.05; *n* = 3) protection of the RPE cells against the oxidative stress, with a least cell death ratio at 100 μM SAC.

### 2.2. Measurement of MCP-1 Protein Using ELISA

In Figure 2, when comparing RPE cells subjected to oxidative stress induced by 500 μM H_2_O_2_ with the ones incubated with DMEM (293.06 ± 2.83; control; *n* = 3), oxidative stress (423.94 ± 11.12; *n* = 3) led to significant (* *p* < 0.001) elevated MCP-1 inflammatory biomarker protein levels. This elevation was significantly († *p* < 0.001) attenuated by 15 min of pretreatment with 25 (280.52 ± 4.66; *n* = 3), 50 (284.80 ± 1.74; *n* = 3), and 100 μM SAC (241.47 ± 6.79; *n* = 3).

### 2.3. Measurement of PKM2 Protein in pRPE via Western Blotting

In Figure 3, the left-hand side blotting showed the expression of PKM2 protein levels of the cells incubated for 24 h in H_2_O_2_ (500 μM) with or without the pre-administration of various concentrations of SAC. RPE cells incubated with DMEM (normal control) were also analyzed. When comparing RPE cells subject to oxidative stress induced by 500 μM H_2_O_2_ with those incubated in DMEM (0.98 ± 0.02; control; *n* = 4), oxidative stress (1.45 ± 0.06) resulted in a significant (* *p* < 0.05) upregulation of PKM2 protein levels. This elevation was dose-dependently ameliorated by 15 min of pretreatment with 50 (1.38 ± 0.22; *n* = 4) and 100 μM SAC (1.00 ± 0.14; *n* = 4), but not at the 25 μM concentration (1.58 ± 0.09; *n* = 4). The most significant († *p* < 0.05) effect was seen at 100 μM SAC.

### 2.4. Electroretinogram (ERG): The Effect of Post-Ischemic Administration of SAC on Retinal I/R

As shown in Figure 4A, there was a considerably reduced ERG b-wave amplitude following pressure-induced retinal ischemia and post-ischemic administration of the vehicle, which is represented as the I/R + Vehicle group. Based on Figure 4B, there was a significant (*** *p* < 0.05) reduction in the b-wave ratio in the I/R + Vehicle group (0.20 ± 0.04; *n* = 8) at 1 day following ischemia, relative to the normal group (1; *n* = 8). Importantly, significant († *p* < 0.05) attenuation of this reduction was shown with post-ischemic administration of SAC at 100 μM (I/R + SAC; 0.43 ± 0.08; *n* = 7).

### 2.5. Fluorogold Labeling

The density of RGCs calculations is shown in Figure 5A, which was divided into the normal retina (left image), the post-ischemic vehicle-treated retina (middle image), and the retina with post-ischemic administration of 100 μM SAC (right image). In Figure 5B, there was a significant (* *p* < 0.05) difference between the normal group (366.78 ± 5.95; *n* = 3) and the I/R + Vehicle group (141.00 ± 9.92; I/R + Vehicle, *n* = 5). Further analysis also revealed a significant difference († *p* < 0.05) between the I/R + Vehicle group and the I/R + 100 μM SAC group (231.70 ± 46.65; I/R + SAC; *n* = 5), which greatly demonstrates the neuroprotective effects of SAC against I/R.

## 3. Discussion

As mentioned previously, it is well known that ischemia plays an important role in diabetic retinopathy, neovascular AMD, CRAO/BRAO, and CRVO/BRVO. More specifically, over the last two decades, anti-VEGF antibody agents and various steroids have been shown to be clinically useful in many of the above-mentioned situations; however, these approaches are not completely effective [2]. Disappointingly, poor visual outcomes have occurred among some patients after anti-VEGF or steroid treatment, even though ocular hemorrhage and macular edema have been successfully controlled [1,12]. New treatments involved in tackling different aspects of the problem, such as upstream inhibitors like PKM2 that further inhibit the downstream VEGF biomarker, and also the usage of anti-inflammatory agents that downregulate the inflammatory biomarker MCP-1 are currently under investigation. These are required in order to effectively treat ischemic retinal disorders when anti-VEGF or steroid treatment fails.

An investigation has been conducted into whether SAC has an anti-oxidative effect on the H_2_O_2_-induced oxidative stress cells, such as RPEs. Moreover, oxidative stress is widely accepted to be related to the ischemic cascade’s key component. Clinically important, the present results of an anti-oxidant SAC might provide an alternative way in the prevention and treatment of ischemic-related ocular disease. On macrophages and the vascular endothelium, SAC is reported to possess a potent anti-oxidant effect, namely scavenging ROS such as superoxides, hydrogen peroxide, and hydroxyl radicals [13]. A recent study has shown that SAC was able to protect against retinal ischemia (not an uncommon cause of visual impairment) via inhibiting the upregulation of VEGF, MMP-9, and HIF-1α [11]. It is therefore vital to study the novel protective mechanisms of SAC, namely in the inhibition of both the upstream biomarker PKM2 and MCP-1, which are known to partake in ischemia-related neovascularization through the recruitment of macrophages that upregulate TNF-alpha along with downstream biomarker VEGF inductions [14]. The present results might be also helpful in finding out novel ways of preventing the development and further deterioration of retinal-ischemic related disorders, such as CRVO, BRVO, CRAO, BRAO, DR, normal-tension glaucoma, and neovascular AMD. These conditions are known to cause a serious array of complications; thus, the proper and timely treatment of retinal ischemia is important in achieving improved patient visual outcomes.

Recent research has also demonstrated that the pig’s immune system is similar to the human’s [15]. The pig eye was therefore presently utilized due to its genetic resemblance to the human eye [15]. The pig comprises similar structural and biological processes to humans, and slaughtered pig eyes purchased from a local monger can be used without any ethical concerns. Thus, the pig RPEs appear to be the appropriate cells for studying the ischemic-related disorders. The human RPE cell line was therefore replaced with self-extracted pig RPE cells. Nevertheless, the self-extracted pig RPE cells compared with commercialized human cell lines is a weaker representation of the human biological milieu. From an evolutionary perspective, a divergent evolution caused the pigs to evolve differently from humans. However, both mammals might share a common ancestor approximately 70 million years ago [16]. Despite the multiple advantages of the current pig eye RPE model in mimicking retinal ischemia, it cannot be generalized to all features of the humans RPEs. Comparative studies or articles on human RPEs will likely be required to be further carried out. In other words, care must be taken in analyzing results as animal cells do not accurately represent human cells every time. In addition, it should be considered that the culture of RPE and H_2_O_2_ would not be the most likely pathological event, especially after isolating RPE cells only. This is due to the neuroretina being more likely sensitive than RPE cells; thus, isolating and exposing them to an H_2_O_2_ insult may not fully mimic in vivo pathological findings in the neuroretina. Nevertheless, it should be acknowledged that there are no perfect models for retinal ischemia, as this is a complex neurodegenerative disorder; for example, neovascular AMD is not yet fully understood. The present results help us understand the mechanisms involved when RPE cells are subjected to oxidative stress.

As discussed in the previous publications, inflammation also played a key role in the development of retinal ischemia-related ocular disorders (e.g., AMD), which includes the proven elevation of the inflammatory biomarkers, namely Cox-2 [17], MCP-1 [18], and iNOS [19]. In the present study, the RPE pig eye model results demonstrate that 500 μM H_2_O_2_-induced oxidative stress significantly stimulated cellular death (Figure 1B,F), relative to the control group of RPE cells cultured in DMEM (Figure 1A,F). In other words, H_2_O_2_ (500 μM) caused free radical formation in cultured RPE cells that significantly reduced the cell viability. As shown in Figure 1, the 15 min pre-treatment of various concentrations of SAC dose-dependently and significantly attenuated cellular death, especially at 100 μM († *p* < 0.05; Figure 1E). In contrast, the least effect was seen at 25 μM (Figure 1C). In summary, the results have shown that SAC is able to counteract the toxic effects of H_2_O_2_ with a dose-dependent and significant protective effect (Figure 1C–F). Thus, the results imply that SAC acts as an anti-oxidant to possibly prevent the development of oxidative stress-related ischemic disorders.

Of clinical novelty, the therapeutic mechanisms of SAC can be explained through the downregulation of MCP-1 and PKM2 secretion (Figure 2 and Figure 3). Specifically, the Western blot assay demonstrated that 100 μM SAC seems to be protective against H_2_O_2_-stimulated oxidative stress through the downregulation of ischemia-related factor PKM2, and it confirms the anti-ischemic role of SAC. Similarly, the ELISA study proves that 100 μM SAC appears to attenuate H_2_O_2_-induced oxidative stress via the downregulation of the inflammatory biomarker MCP-1, and it supports an anti-inflammatory role for SAC. This can be attributed to MCP-1′s ability to attract and recruit macrophages [14], which activates the downstream proteins TNF-alpha and VEGF, resulting in ischemia-induced neovascularization [14]. The downregulation of these biomarkers is said to lower the rate of neovascularization, vascular permeability, vascular dysfunction, and further inflammatory complications [9], which is associated with numerous retinal-related disorders. Consistently increased levels of HIF-1α, VEGF, p38 mitogen-activated protein kinase (MAPK), and MMP-9 [11], as well as MCP-1 (Figure 2) and PKM2 (Figure 3), were detected during the previous and present studies. These are assumed to be linked in some way to retinal ischemia-associated diseases, such as neovascular AMD, CRAO/BRAO and CRVO/BRVO. In such circumstances, based on the present results, SAC might provide an alternative way to deal with oxidative stress or retinal ischemia-related vision threatening retinal disorders.

Previous studies have reported that that pre-administered SAC could attenuate retinal ischemia- or kainate excitotoxicity-induced reduction in ERG b-wave ratios [11,20]. As for the present ERG data (Figure 4), they also show that I/R causes decreased neuronal physiological function via smaller b-wave ratios relative to the normal/control group. Importantly, this study proved that post-administration of 100 μM SAC (I/R + SAC) counteracted the ischemia-linked decrease in the ERG b-wave. Moreover, the fluorogold retrograde labeling RGCs shows a decreased number of RGCs in the in vivo retinal ischemic model. In this case, it is well known that these cells are responsible for the transmission of visual information from the retina to the central nervous system; thus, the death or absence of these cells can result in permanent vision loss. RGC qualitative and quantitative results have illustrated that the post-ischemic administration of 100 μM SAC (I/R + SAC) significantly counteracted the ischemia-associated RGC cell number reduction (Figure 5A,B) when compared with the I/R and vehicle treatment group.

In this case, a retinal ischemic model was employed through elevated IOP to gain understanding into the medicinal properties of SAC against I/R injury. Although high IOP I/R serves as a good framework for understanding the pathology of retinal ischemia and the subsequent recovery mechanism, a cautionary note should be taken for extrapolating the results to chronic conditions, as it is only a simulation and a presumed acute ischemia animal model [21]. Also, extra-surgical variables (e.g., well-anesthetized for intravitreous injection and 1 h HIOP procedure) before surgery can potentially affect drug potency and mortality; thus, the careful addressing of these factors enables the establishment of a well-controlled and reproducible model of I/R injury [21]. These are potential limitations and weaknesses that can be associated with the current study. Given the model’s well-established anatomical accuracy, experimental replicability, and technical ease, this still remains a promising tool for looking at neuronal conditions associated with inflammation, ischemia, and hemorrhaging of the retina [7,21].

Overall, the cell culture studies, electrophysiology, and fluorogold retrograde labeling immunohistochemistry studies have not only demonstrated the harmful effects of oxidative stress and I/R on the retina, but have also shown SAC’s ability to attenuate these ischemic-associated detrimental alterations. These include RPE cell death, the upregulation of ischemic and inflammatory biomarkers like PKM2 and MCP-1, the lowering of ERG b-wave amplitude, and the decrease in RGC cell count. Based on the present data, the results imply that both pre-administered and post-administered SAC (100 μM) might act as an alternative and complementary treatment for ischemic-related disorders using presently proven novel mechanisms, namely MCP-1 and PKM2 downregulation, when conventional treatments like anti-VEGF and steroidal intravitreal injections fail.

## 4. Materials and Methods 

### 4.1. In Vitro Studies and Protocol to Isolate RPE Cells

The RPE extraction protocol was carried out according to the publication of Toops et al. (2014) with some modifications [16]. A total of 15 dead pig eyes was purchased from a monger at Taipei in Taiwan. Eyeballs were waste products from already-slaughtered pigs. This conforms to the animal ethical policy, as no injury was given to the pigs. Immediately after purchase, the pig eyes were stored at 4 °C. The pig eye’s extraocular muscles, cornea, and lens were dissected. The eyeball was then incubated for 10 min in 1% povidone-iodine solution on cold ice. The eyeball was further washed with sterile distilled water five times and then soaked for five minutes in 1000 U/mL of penicillin/streptomycin (10,000 U/mL diluted 1:10 in sterile distilled water) on ice. Intraocular tissues (e.g., vitreous at the ora serrata) were isolated with a spatula. The retina was gently separated further and retrieved from the eye cup. The eye cups (posterior pole) were instilled with 1 mL of 0.5% warmed trypsin together with 5.3 mM EDTA in HBSS without calcium and magnesium. It was then placed in a 5% CO_2_ incubator for 30 min at 33 °C. Consequently, the 1 mL pipet was used to separate the RPE cells by repeated pipetting. The solution containing the RPE cells was retrieved in tubes for centrifugation. This was processed at 1000× *g* for 10 min. The RPEs were then precipitated at the bottom of the tubes. The precipitated RPE cells were placed in 10% FBS in Dulbecco’s Modified Eagle Medium (DMEM) with 4.5 g/L glucose, L-glutamine and sodium pyruvate, 1% NEAA, and 1% penicillin/streptomycin. The cells were then incubated in a 5% CO_2_ incubator at 33 °C. 

### 4.2. Cell Culture

The RPE cells together with 1.5 mL of DMEM were added into 6-well plates (cell concentration: 3 × 10^5^/well). The cells were then cultured in DMEM medium at 33 °C for a day before the inclusion of H_2_O_2_ with or without SAC. The groups were divided into either the normal control group (incubation of DMEM), experimental control group (incubation of H_2_O_2_ to stimulate an oxidative stress), or treatment group (H_2_O_2_ + SAC). SAC was added to cultured RPE cells in each defined group 15 min before adding H_2_O_2_ (500 μM). After 24 h of incubation, 10 μg/mL propidium iodide (PI) was added to stain the RPE cells in the dark under epifluorescence microscopy. The images of the stained (dead) and unstained cells (living) were mingled utilizing Adobe Photoshop CS5. The cells were further calculated by a hemocytometer to assess the cell death rate. The effect of SAC on an oxidative stress was analyzed by comparing the cell death rates between various mentioned groups. Each experiment was carried out in triplicate. 

### 4.3. ELISA for MCP-1

ELISA kits for RPE cell supernatants were purchased from R&D Systems (Minneapolis, MN, USA) and used to detect MCP-1 (catalog no. DCP00) based on the manufacturer’s instructions. All assays were performed in triplicate, and the average optical density of each sample was measured via a Synergy H1 Multi-Mode reader BioTek Instrument. MCP-1 [22] levels were expressed as pg/mL and calculated based on the manufacturer’s protocol.

### 4.4. Gel Electrophoresis and Western Blotting

As for the Western blot assay, the RPE cells were separated by treating with trypsin–ethylenediaminetetraacetic acid (EDTA) from the culture plate at 2 h after incubation of DMEM or 500 μM H_2_O_2_ with or without SAC (25, 50, or 100 μM). The cells were collected following centrifugation and rinsed 3 times in cold phosphate-buffered saline (PBS). Proteins were retrieved by sitting at 4 °C in ice-cold mammalian protein extraction reagent (HyCell Biotechnology Inc., Logan, UT, USA) for 30 min and vortexing every 10 min. The samples were then centrifuged at 15,000× *g* for 10 min.

Equal amounts of denatured proteins (40 μg/20 μL/well) were treated with a sodium dodecyl sulfate polyacrylamide gel electrophoresis (SDS-PAGE; Bio-Rad, Hercules, CA, USA) as described previously [12]. Then, SDS-PAGE was moved to a polyvinylidene fluoride (PVDF) membrane and a PVDF membrane blocked with 5% nonfat milk in PBST solution for 1 h. The membranes were then incubated 16 h at 4 °C with different primary antibodies: rabbit polyclonal anti-β-actin antibody (1:2000; GeneTex, Inc., Irvine, CA, USA) and rabbit monoclonal antibody PKM2 (1:1000; Cell Signaling Technology Inc., Cambridge, UK). The blots were further incubated with a relevant secondary antibody, horseradish peroxidase-conjugated goat anti-rabbit IgG (1:2000; GeneTex, Inc., Irvine, CA, USA), at 37 °C for 1.5 h. Immunoblot images were captured and analyzed with UVP ChemStudio PLUS (Analytik Jena GmbH, Jena, Germany). The densitometry of immunoblot images was performed with ImageJ version 1.53t [12].

### 4.5. In Vivo Studies

#### Anesthesia and Euthanasia

Intraperitoneal injection was utilized to anesthetize the rats. In this case, 100 mg/kg ketamine (Pfizer, New York City, NY, USA) and 5 mg/kg xylazine (Sigma-Aldrich, St. Louis, MO, USA) were infused into the rats. Then, infusion of 140 mg/kg sodium pentobarbital (SCI Pharmtech, Taoyuan, Taiwan) was provided to euthanize the animal. 

### 4.6. Animals

The following research has strictly obeyed the rules set by the ARVO Statement for the Use of Animals in Ophthalmology and Vision Research. In addition, the agreement to perform the research was received from the institutional review board of Cheng-Hsin General Hospital (Taipei, Taiwan; Approval No: CHIACUC 106-09). In terms of the animals, six-week-old Wistar rats (BioLasco, Taipei City, Taiwan) were ordered and raised at a humidity of 40–60% and a temperature of 19–23 °C. The rats were reared on a 12 h light/dark cycle together with 12~15 air refreshments per hour and were provided with food and water at liberty. Then, the animals were randomly allocated into normal, post-ischemic vehicle (I/R + Vehicle), and post-ischemic treatment groups (I/R + SAC).

### 4.7. Retrograde Labeling of RGCs

Following anesthetics administration, the scalp of the rat was incised with a 2 cm cut along with 2 small holes drilled into the skull on both sides behind the bregma and 1.5 mm lateral to the midline [12]. Through a micropipette, 2 μL of 5% fluorogold (Sigma-Aldrich, St. Louis, MO, USA) was injected at depths of 3.8, 4.0, and 4.2 mm beneath the cranium. Three days following RGCs’ retrograde immunolabeling, pressure-induced retinal ischemia was performed on the animals’ right eyes, with the adjacent eyes being the untreated ones. The twelve o’clock location of the rat’s eye was highlighted using a suture to carry out orientation; then, the eyes were removed from their sockets. Following this, the retina was carefully retrieved, fixated, dissected, and processed [11]. Then, the retina was placed onto a slide and divided into 4 quadrants. Then, each retinal quadrant was split into 3 zones (i.e., central, middle, and peripheral) at distances of approximately 1, 2, and 3 mm from the optic disc, respectively [20]. Inside the aforementioned zones, and with the assistance of a microscope, 6 fields of 0.430 × 0.285 mm^2^ each along the medial line were tallied. In this case, a total of 72 fields of the entire retina was tallied [20]. The average RGC density was evaluated by obtaining the ratio of the total RGC number against the total retinal area [20]. 

### 4.8. Ischemia Induction

The rats were anesthetized and placed onto a stereotaxic frame. Retinal ischemia was induced via increasing intraocular pressure (IOP) to 120 mmHg for 1 h in the rat’s eye via a 30 G needle that was linked to a 0.9% 500 mL physiological saline bottle and infused into the anterior chamber. Specifically, the bottle was placed at 163 cm above the eye to the apex of the saline reservoir, which is said to be equivalent to 120 mmHg [20]. In this case, manifestation of retinal whitening was noted. The animals were placed on 37 °C heating pads during the ischemia induction. This ischemia is followed by reperfusion (I/R) for 1 day. Whether post-ischemic “SAC” might attenuate retinal ischemic injury and its underlying mechanisms were evaluated using an electroretinogram (ERG) and fluorogold retrograde labeling as defined.

### 4.9. Drug Administration

Drug administration involved the post-ischemic administration of SAC (100 μM) or a vehicle (saline; control). The ischemic eye of each test rat was treated without (normal) or with a single intravitreous injection of 5 μL of the defined test compounds; the fellow normal eye was untreated.

### 4.10. ERG

Flash ERG was conducted on all rats one day after I/R and treatment with SAC or a vehicle. The rats were adapted in the dark for a total of 8 h, then anesthetized prior to ERG measurements with pupil dilation with 1% tropicamide and 2.5% phenylephrine (Akorn, Inc., Lake Forest, IL, USA); then, the cornea was applied with 0.5% proparacaine anesthetics (Alcon, ZG, Vernier-Geneva, Switzerland). The rat’s eye was stimulated at a frequency of 0.5 Hz through a torch placed 20 mm in front of the eyes. Fifteen consecutive data points at 10 kH were retrieved at an interval of 2 s. The responses were recorded and totaled using an amplifier P511/regulated power supply 107/stimulator PS22 (Grass-Telefactor; AstroNova, Brossard, QC, Canada). The b-wave ratio regarding the amplitude of the treated ischemic eye versus untreated fellow normal eye was compared [12]. As for the exclusion criteria, they excluded the rats’ b-wave ratios being above 125% and below 75%.

### 4.11. Statistical Analysis

Unpaired Student’s *t*-tests were used to compare 2 independent groups. The results are the mean ± standard error. A probability of less than 0.05 (*p* < 0.05) was considered significant.

## 5. Conclusions

The present cellular H_2_O_2_ inclusion assay and animal HIOP induction model results have proven that both pre- and post-administered SAC (100 μM) possess beneficial therapeutic effects in terms of treating retinal ischemic injury and oxidative stress. Furthermore, as revealed by the in vitro analysis and in vivo studies, SAC might protect against retinal ischemia by acting as an anti-oxidant and inhibiting the upregulation of upstream biomarkers MCP-1 and PKM2, as presently demonstrated in RPE cells subjected to 500 μM H_2_O_2_-induced oxidative stress. In addition, SAC attenuated the I/R injury-associated reduction of ERG b-wave amplitudes and density of RGCs. Overall, SAC has been proven to be able to protect against retinal ischemia and oxidative stress through its free radical scavenging, anti-ischemia, possible anti-angiogenesis, and anti-inflammation properties, as well as its downregulation of PKM2 and MCP-1 biomarker levels.

## Figures and Tables

**Figure 1 ijms-25-01349-f001:**
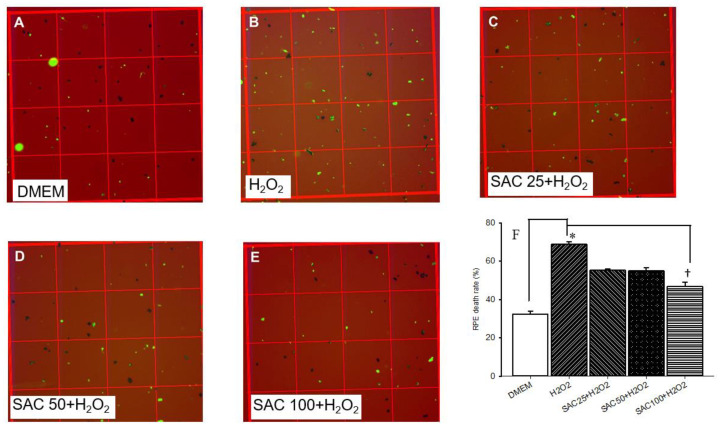
Optical microscopy was utilized to observe cell density as a representation of the cell death ratio. In the control group (**A**), for RPE cells cultured in DMEM, there were numerous RPE cells with the least cell death ratio. Furthermore, oxidative stress induced by incubation of the RPE cells with 500 μM H_2_O_2_ (**B**) for 24 h drastically reduced the number of RPE cells, which is associated with a significantly greater cell death ratio. In contrast with (**B**), (**C**–**E**) indicate that 15 min of pre-incubation with 25, 50, and 100 μM SAC resulted in a dose-dependent and significant protection of the RPE cells against the oxidative stress, which was particularly associated with the significantly smallest cell death ratio at 100 μM SAC. In (**B**,**F**), compared with the RPE cells cultured in DMEM (control; **A**,**F**), a 24 h incubation of RPE cells with 500 μM H_2_O_2_ significantly (* *p* < 0.05) increased the cell death ratio or reduced the cell viability. In contrast, 15 min of pre-incubation with 25, 50, or 100 μM of SAC dose-dependently and significantly (at 100 μM; † *p* < 0.05) ameliorated the H_2_O_2_-induced increase in the cell death ratio. RPE living cells: black; PI-stained dead cells: green. Results are represented as the mean ± SD (*n* = 3).

**Figure 2 ijms-25-01349-f002:**
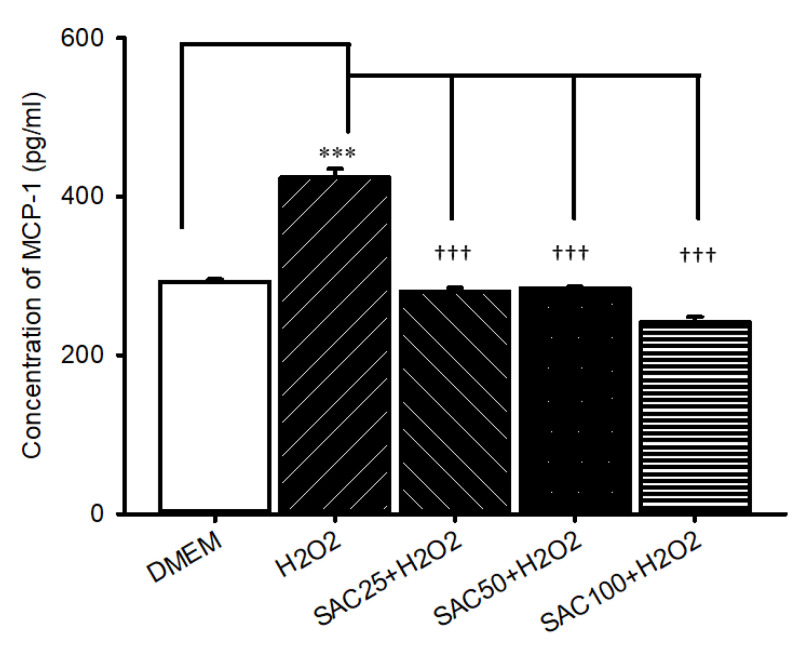
Analysis of the expression of MCP-1 protein levels by ELISA. Cells were evaluated twenty-four hours after 500 μM H_2_O_2_-induced oxidative stress relative to cells incubated in DMEM (control). Oxidative stress resulted in a significant (*** *p* < 0.001) upregulation of MCP-1 protein levels. This significant elevation was significantly (††† *p* < 0.001) attenuated by 15 min of pretreatment with 25, 50, and 100 μM SAC. Results are displayed as the mean ± SD (*n* = 3).

**Figure 3 ijms-25-01349-f003:**
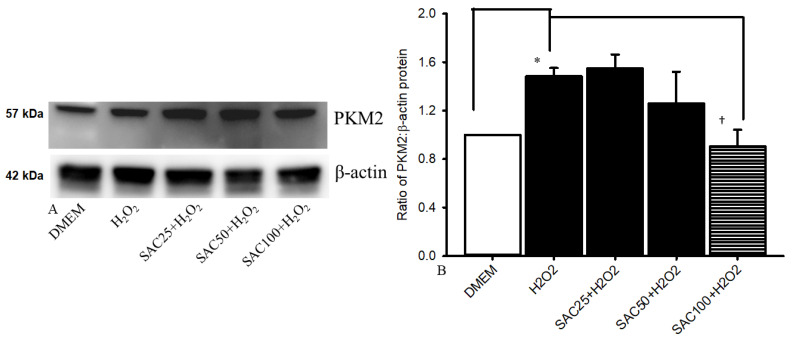
Analysis of the expression of PKM2 protein via Western blotting. The left image showed the blots for expressions of β-actin (lower row) and PKM2 protein (upper row). Column 1 in the right bar chart showed the cells that were cultured in DMEM; column 2 demonstrated the cells that were incubated in H_2_O_2_; columns 3–5 revealed the H_2_O_2_-injured cells that were pre-administrated with 25/50/100 μM of SAC for 15 min. Twenty-four hours after 500 μM H_2_O_2_-induced oxidative stress in comparison with the cells incubated in DMEM (control), the analysis of cells revealed that oxidative stress resulted in a significant (* *p* < 0.05) upregulation of PKM2 protein levels. This elevation was significantly († *p* < 0.05) mitigated by 15 min of pretreatment with 100 μM SAC. Results are expressed as the mean ± SD (*n* = 4).

**Figure 4 ijms-25-01349-f004:**
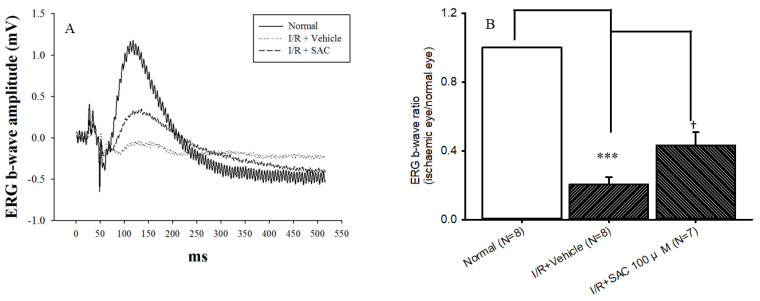
Electroretinogram (ERG): the effect of post-ischemic administration of SAC on retinal ischemia plus reperfusion (I/R). (**A**) When compared with the normal retina (control), there was a considerable reduction in the amplitudes of the ERG b-wave following pressure-induced retinal ischemia and post-ischemic administration of the vehicle in a representative animal. Improvement of this reduction was shown with the post-ischemic administration of SAC (100 μM, I/R + SAC). (**B**) In contrast with the normal group (*n* = 8), there was a significant (***, *p* < 0.05) reduction in the b-wave ratio in the I/R + Vehicle group (*n* = 8) at 1 day following I/R. Significant (†, *p* < 0.05) amelioration of this ischemia-induced decrease was achieved with post-ischemic administration of 100 μM SAC (I/R + SAC; *n* = 7). Results are illustrated as the mean ± SD.

**Figure 5 ijms-25-01349-f005:**
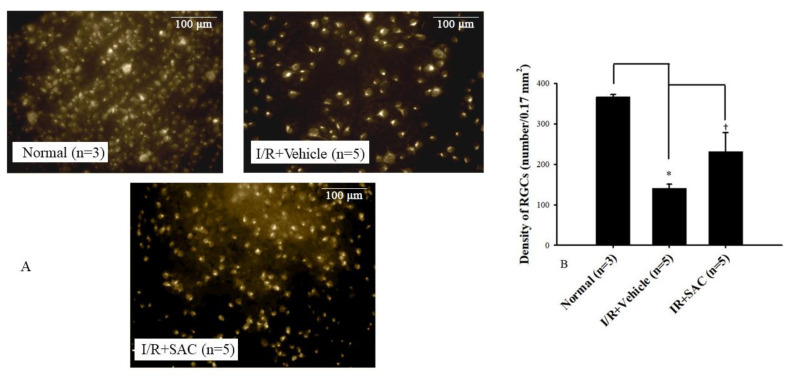
Fluorogold labeling. (**A**) The microscopic images demonstrate the cell density of retinal ganglion cells (RGCs) for the control retina (normal), the ischemic retina post-ischemia administrated with vehicle (I/R + Vehicle), and the ischemic retina post-ischemia administrated with 100 μM SAC (I/R + SAC) for one day. Scale bars (white) = 100 μm. (**B**) Each bar represents the mean ± SEM. There was a significant difference (* *p* < 0.05) between the normal group (*n* = 3) and the I/R + Vehicle group (*n* = 5). A significant difference († *p* < 0.05) was also demonstrated between the I/R + Vehicle group and the I/R + SAC group (*n* = 5).

## Data Availability

Data are available from the authors upon request.

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
