# Peer review of "The Effect of S-Allyl L-Cysteine on Retinal Ischemia: The Contributions of MCP-1 and PKM2 in the Underlying Medicinal Properties"

_ijms, 2024, doi:10.3390/ijms25021349_

Round 1

Reviewer 1 Report

Comments and Suggestions for Authors

The study by Chao and cols., explores how S-allyl L-cysteine protects against retinal ischemia. The authors demonstrated dose-dependent neuroprotective effects of S-allyl L-cysteine in RPE cells subjected to oxidative stress, with higher efficacy at 100 μM. In vivo, S-allyl L-cysteine preserved retinal function and RGCs in a model of ischemia/reperfusion. The research follows the standards for the ethics of experimentation and research integrity. The material and methods are not described in sufficient detail (e.g., the number of pig eyes). The number of samples/assay are detailed, but the statistical analyses are appropriate. The injury model has been widely used as an ischemia/reperfusion model and is sometimes reported as an acute model for glaucoma. Although the authors examine the retinal function and the RGCs viability the analysis of the protective effects of SAC in rat retina is a bit poor and lacks other confirmation. Below there are some points to consider and a few suggestions to further improve the manuscript.

1. In the "Ischemia induction" section of the Material and Methods, provide additional details about the 0.9% saline bottle volume and its height relative to the rat eye due to variations in IOP regarding other publications.

2. In the "In vitro studies" section of the Material and Methods, please provide how many pig eyes were used in this study.

3. For the evaluation of RPE cell survival after pre-administration of SAC. Have the authors evaluated the effect of DMEM + SAC at 25, 50, and 100 uM (without H2O2) to establish basal levels for survival, MCP1, and PKM2 (Fig. 1, 2, 3)?

4. The protective effect at 100 uM did not reach a plateau or reverse effect. Could it be improved? Have the authors evaluated higher concentrations of SAC to verify that 100uM is the optimal concentration?

5. This study analyzed the protein level of MCP1 and PKM2 in ex vivo RPE cells to establish the most effective concentration. However, RPE cells and the H2O2 challenge are very different from neuronal cells and ischemia/reperfusion injury, respectively. Have the authors verified whether those levels were also reduced in the rat retinas when administrating SAC 100uM after I/R?

6. In the "Fluorogold injections" section on Page 5 (line 3), provide injection coordinates (from bregma) and specify whether injections were bilateral and targeting both superior colliculus (e.g., PMID: 25482219).

7. Fluorogold images (Fig. 5A). i/ The resolution of the Fluorogold images is poor, could the author increase the resolution? ii/ The images for Fluorogold-traced RGCs do not seem to belong to comparable areas. The RGC somas in the I/R+Vehicle are bigger than those for Normal or I/R+SAC, which resembles a more peripheral area where fewer RGCs lay. Please, ensure that images represent comparable areas.

8. In the retrograde labeling section (Material and methods, 2.10, page 5), briefly explain if RGC density calculation involved quantifying whole retinas how many areas were analyzed, and if they were acquired in similar regions.

9. Fluorogold signal and RGC survival interpretation. The authors retrogradely labeled the RGCs by applying Fluorogold 3 days before inducing I/R. But, i/ if RGCs were properly labeled before dying, the phagocytic microglial cells would become transcellularly labeled as well (after phagocyting any RGC with Fluorogold) as previously reported (e.g., PMID: 1383017, 29121969). However, phagocytic Microglial cells are not distinguishable in these images. ii/ Other interpretations could suggest axonal transport impairment (not specifically RGC death yet). If RGCs were not fully labeled at the time that I/R was performed (note that other authors recommend 5-7days for tracing RGCs from the brain in rats, PMID: 3181354, 25482219) and the axonal transport is affected, the tracer may not reach the soma. Although the Normal group (Fig 5A) presents reliable labeling, it has an additional day for the tracing (the day after I/R). Thus, how do the authors interpret the absence of phagocytic Microglial cells if RGC death has occurred after tracing?

10. Have the authors evaluated other degenerative events? Such as inflammation or the immune response? E.g. Examining microglial cell proliferation, activation... (Iba1, CD68 antibodies); apoptosis (Casp3, TUNEL...)

11. Considering a translational plan. Have the author tested the same paradigm in rats but instead of injecting iv S-allyl L-cysteine, using a diet enriched e.g., with garlic? Could it be considered a preventive strategy?

12. Have the authors tested if pre-administration of SAC (e.g. during 2 weeks) improves the results obtained after post-injury treatment?

13. The study only includes one-time point (1 day post-I/R). To interpret the relevance of this treatment, how long does the RGC survival last? And could several administrations maintain this protective effect?

14. Discussion (page 10, lines 3-6). Despite the similarities between the pig and human retinas. The culture RPE and H2O2 insult would not be a most likely pathological event, especially after isolating RPE cells only. The neuroretina is more likely sensitive than RPE cells. Please, acknowledge that isolating RPE cells and subjecting them to H2O2 insult may not fully replicate in vivo pathological events in the neuroretina.

Minor comments:

15. Please check the “alpha” symbol for all HIF-1a calls in the text. E.g., Page 2, lines 9, and 11.

16. In material and methods. Since the “Retrograde labeling of RGCs” was performed before the “Ischemia induction”, I would recommend rearranging the order of sections. “Retrograde labeling of RGCs" would precede "Ischemia induction".

17. In material and methods, in the retrograde labeling (section 2.10). Page 5, line 4: “Three days following RGCs’ retrograde immunolabeling,”. Fluorogold is not an immunolabeling method, please replace it with retrograde ‘labeling.’

18. Please ensure that the final version of the manuscript has included the comments regarding Author contributions, funding, etc... (page 11) before submitting.

Comments on the Quality of English Language

I did not detect major errors

Reviewer 2 Report

Comments and Suggestions for Authors

The current study examined the role of S-allyl L-cysteine (SAC) and its associated therapeutic mechanism in a rat model of ischemic retina. Through electrophysiology, fluorogold-labelling, ELISA, and western blot, the authors showed that the neuroprotective effect of pre-administered SAC was dose-dependent, and that SAC downregulated novel up-stream ischemia/angiogenesis related factor PKM2 and inflammatory biomarker MCP-1. Moreover, by utilizing the electro-physiology measurement, the authors demonstrated that post-administration of SAC in Wistar rat’s ischemic retina attenuated ischemia induced decrease in the ERG b-wave amplitude and reduction in the fluorogold labelling RGC number. The authors suggested that SAC could protect against retinal ischemia through its antioxidative, anti-angiogenic, anti-inflammatory, and neuroprotective properties. Overall, the experiments are properly performed and their in vivo observation is clinically sound. However, some concerns listed below limit the clear narrative of the current study. 

1.     The abstract requires reorganization to succinctly and clearly present the major findings. Excessive detailed information regarding the methodology employed in the current study is unnecessary for the abstract. Please revise it to enhance its presentation for readers.

2.     In the Figures for western blot, the molecular weight should be labelled for the proteins evaluated.

3.     Is there any side effect of the S-allyl L-cysteine? The authors should provide experimental results to evaluate the potential side effects of this drug on vital organs in normal animals.

4.     In the Discussion Section, it is important for the reader to know what is limitation/weakness of the current manuscript. The authors should comment, if possible, on potential limitations and weaknesses for this manuscript, before concluding the discussion.

5.     The manuscript's formatting should adhere to the specifications outlined by this journal.

6.     Proofreading is required, including addressing issues such as improper spacing between words.

Comments on the Quality of English Language

Editing of English language is required before consideration of publication.

Round 2

Reviewer 2 Report

Comments and Suggestions for Authors

The authors have properly addressed my concerns in the revised manuscript. I have no further comments.